# The Thermodynamically Expensive Contribution of Three Calcium Sources to Somatic Release of Serotonin

**DOI:** 10.3390/ijms23031495

**Published:** 2022-01-28

**Authors:** Francisco F. De-Miguel

**Affiliations:** Instituto de Fisiología Celular-Neurociencias, Universidad Nacional Autónoma de México, Mexico City 04510, Mexico; ffernand@ifc.unam.mx

**Keywords:** calcium channels, somatic exocytosis, serotonin, extrasynaptic release, thermodynamic efficiency, leech

## Abstract

The soma, dendrites and axon of neurons may display calcium-dependent release of transmitters and peptides. Such release is named extrasynaptic for occurring in absence of synaptic structures. This review describes the cooperative actions of three calcium sources on somatic exocytosis. Emphasis is given to the somatic release of serotonin by the classical leech Retzius neuron, which has allowed detailed studies on the fine steps from excitation to exocytosis. Trains of action potentials induce transmembrane calcium entry through L-type channels. For action potential frequencies above 5 Hz, summation of calcium transients on individual action potentials activates the second calcium source: ryanodine receptors produce calcium-induced calcium release. The resulting calcium tsunami activates mitochondrial ATP synthesis to fuel transport of vesicles to the plasma membrane. Serotonin that is released maintains a large-scale exocytosis by activating the third calcium source: serotonin autoreceptors coupled to phospholipase C promote IP3 production. Activated IP3 receptors in peripheral endoplasmic reticulum release calcium that promotes vesicle fusion. The Swiss-clock workings of the machinery for somatic exocytosis has a striking disadvantage. The essential calcium-releasing endoplasmic reticulum near the plasma membrane hinders the vesicle transport, drastically reducing the thermodynamic efficiency of the ATP expenses and elevating the energy cost of release.

## 1. Introduction

Streams of action potentials or long depolarizations may evoke calcium-dependent exocytosis of transmitters and peptides from the soma, dendrites and axon of neurons in the absence of pre- and postsynaptic structures [1,2,3,4,5,6,7,8]. In contrast to the fast and localized release from synapses, extrasynaptic release is slow and diffuse, producing effects by means of volume transmission of signaling substances [9]. Synaptic release contributes to rapid communication by neuronal circuits. By contrast, substances that are released extrasynaptically reach targets at variable distances to modulate the workings of populations of neurons, glia and blood vessels [10,11]. An extensive list of reviews on the effects of extrasynaptic exocytosis can be found in the literature [8,12,13,14,15,16,17,18,19,20].

All peptides and most classical low molecular weight transmitters have been shown to be released extrasynaptically from vesicles, either clear or often dense-core vesicles that rest distantly from the plasma membrane [6]. Therefore, bursts of electrical activity or long depolarizations and calcium entry are needed to mobilize vesicles to the plasma membrane [21,22]. Calcium is not only essential for vesicle fusion, but has intermediate essential roles in the vesicle transport.

This review summarizes experimental evidence about the ways by which calcium flow through transmembrane and intracellular channels contributes in an orderly manner to induce vesicle transport and a large-scale exocytosis that lasts for hundreds of seconds. The backbone for the paper is experiments made on the soma of the classical large serotonergic Retzius neuron of the leech CNS, which allows direct experimentation to analyze the sequence of fine steps that couple electrical activity with exocytosis [22]. Data supporting the generality of our results, obtained from different types of neurons, tissues and animals are mentioned along the text. For the moment it is useful to mention that electrical stimulation of a Retzius neuron with trains of 10 impulses at 20 Hz in an isolated ganglion activates the neuronal circuit that controls crawling locomotor behavior [23]. During crawling, bursts of motoneuron firing produce alternate elongation of front body segments with shortening of rear segments [24,25]. The minute range latency to crawling is due first, to the series of calcium-dependent processes (the matter of this review), that take place between electrical activity and serotonin liberation. Unpublished experiments by Sanchez-Sugía and the author of this review show that the second part of the long latency to behavior reflects the arrival time of released serotonin to distant neuron targets that codify for crawling (Sanchez-Sugía C., and De-Miguel, F.F., in preparation).

## 2. Electrical Activity and Calcium Promote Vesicle Transport and Somatic Exocytosis

Ultrastructural analysis of the soma of Retzius neurons under different stimulation frequencies permits an initial understanding of the role of electrical stimulation and calcium on somatic exocytosis. The electron micrographs in Figure 1 show the effects of electrical stimulation and calcium entry on somatic clusters of serotonin-containing dense core vesicles [26]. The injection of brief depolarizing current pulses through a microelectrode to produce trains of action potentials at 1 Hz trains (Figure 1), a frequency similar to that of spontaneous firing in serotonergic and other monoaminergic neurons [27,28,29,30], fails to promote any vesicle mobilization to the plasma membrane and exocytosis. However, increasing the frequency of the train to 20 Hz (Figure 1B) produces ~50% of the vesicle clusters in electron micrographs to appear closely apposed to the plasma membrane [26]. If a similar 20 Hz train is delivered in the presence of extracellular horseradish peroxidase, gold-coupled anti-peroxidase antibodies recognize peroxidase inside vesicles that underwent the exo-endocytosis cycle [31]. Our laboratory has also found that after 20 Hz stimulation in the presence of extracellular magnesium to block transmembrane calcium entry, vesicles remain at resting positions (Del Pozo-Sánchez, V., Mendez, B. and De-Miguel, F.F., in preparation).

## 3. Calcium Enters the Neuronal Soma through L-Type Channels

The first source of calcium for somatic exocytosis in Retzius neurons is the transmembrane flow through L-type (CaV1) channels [32,33]. L-type channels are well suited for somatic exocytosis in Retzius neurons, dopaminergic retinal neurons, serotonergic neuroepithelial neurons and in certain peptidergic neurons [3,32,33,34,35], for their slow inactivation permits continuous calcium inflow during long depolarizations or trains of action potentials. In dendrites of magnocellular neurons releasing vasopressin and oxytocin and in other neuron types, N type (CaV2.2) channels contribute to exocytosis alone or in collaboration with L-type channels [35,36]. However, inactivation of N-type channels restricts their frequency-responses over long periods of electrical activity. In dendrites of magnocellular neurons, serotonergic neurons and dopamine ganglion neurons of the retina calcium elevations may result from activation of glutamate receptors, usually in the absence of local action potentials [3,37,38].

Evidence on the presence of L-type channels in the plasma membrane of the soma of Retzius neurons came first from loose patch clamp recordings of currents from small areas of plasma membrane [39]. Figure 2 shows that the somatic calcium tail currents are smaller than those in the axon stump, a preferential region for the regeneration of neurites and synapse formation in culture [39]. The kinetics of activation of somatic calcium currents was reconstructed by superposing isolated calcium tail currents recorded from a patch of membrane on increasingly larger depolarization voltage pulses. As shown in Figure 2B, the calcium current inactivates slowly, as expected for L-type currents but not for calcium currents carried by other types of calcium channels.

## 4. High-Frequency Activation of L-Type Channels Promotes Somatic Exocytosis

Somatic exocytosis of serotonin can be evoked in Retzius cells in a frequency-dependent manner by trains of action potentials but not by individual action potentials [32,40]. Exocytosis on stimulation trains of ten impulses can be conveniently visualized from the formation of fluorescent spots of FM1-43 right inside the plasma membrane [32]. Each spot forms upon the accumulation of dye inside a cluster of vesicles that undergo exo-endocytosis cycles. Although 1 Hz stimulation fails to evoke exocytosis, a number of fluorescent spots are produced by constitutive exocytosis. Such staining occurs equally well in the absence of stimulation, on 1 Hz stimulation, or on 20 Hz stimulation under experimental conditions that eliminate calcium entry [32]. Above 5 Hz, the number of fluorescent spots increases with the stimulation frequency [40]. Figure 3 compares the FM1-43 staining patterns of cultured neurons, one stimulated with 1 Hz train (Figure 3A) and another stimulated with 10 Hz train (Figure 3B). The staining pattern produced in culture by 10 Hz stimulation reproduces that of neurons stimulated in the ganglion (Figure 3C), therefore, the culture conditions provide an accurate bioassay to follow the sequence of events that led to serotonin exocytosis [32]. A plateau in the amount of release at 20 Hz in Retzius neurons, in lobster serotonergic neurons and in mammalian peptidergic neurons [40,41,42], suggests a generalized peak of efficiency for the coupling between electrical activity, calcium and exocytosis. For the reasons above, the experiments to be described next were carried out by stimulating neurons with 1 Hz or 20 Hz trains in combination with different experimental manipulations.

Blockade of L-type channels through the incubation of neurons with nimodipine abolishes exocytosis upon 20 Hz stimulation without affecting the constitutive exocytosis (Figure 4A) [32,40]. Subsequent addition of caffeine to such same neurons to promote release of intracellular calcium through ryanodine receptors [43,44,45,46] in the absence of electrical stimulation produced a bulk of exocytosis, seen as an increased number of fluorescent spots (Figure 4B). Such results clarified that calcium entry upon activation of L-type channels promotes both vesicle transport and exocytosis. Moreover, they suggested that calcium-induced calcium release contributes to link excitation and exocytosis.

## 5. Analysis of the Calcium Dynamics

Essential to understanding the translation of the firing frequency into the amount of somatic exocytosis was to image the calcium elevations from the fluorescence of the high affinity Fluo calcium sensors upon different stimulation frequencies. Such measurements showed that the calcium transient on each individual action potential experienced frequency-dependent summation [40]. Figure 5 superimposes the calcium kinetics in response to 1 Hz and 20 Hz trains. Whereas calcium transients produced by each action potential at 1 Hz could be well-resolved, summation at 20 Hz frequency produced a rapid large wave that peaked by 600 ms and decayed exponentially with a ~3 s time constant. The remarkable difference between the peak and the integral of the calcium transient shown in Figure 5B,C indicates that it is the concentration, not the total amount of calcium, that determines the amount of exocytosis [40]. Moreover, the supralinear increase in the amplitude of the calcium signal on 20 Hz stimulation (Figure 5B,C), suggested the recruitment of a second source of calcium. Such observation goes in line with the effects of caffeine inducing exocytosis in the presence of nimodipine (Figure 4).

Figure 6 shows that the fast calcium transient produced by a 20 Hz train acts as a tsunami to invade the whole soma [40]. An additional and unexpected observation shown in Figure 6 is that the fast calcium transient is followed by the appearance of a calcium transient exclusively at the soma shell. Such transient persists for hundreds of seconds in the absence of any stimulation and reflects the time course of somatic exocytosis.

Summarizing these observations, 20 Hz stimulation produces summation of the transmembrane calcium elevation that follows each action potential. Activation of calcium-induced calcium release, the second pool of calcium, rapidly increases the concentration of the calcium wave that invades the soma. A third calcium pool in the periphery correlates with the duration of exocytosis.

## 6. The Fast Calcium Transient Determines the Amount of Release

The possibility of calcium-induced calcium release being the second calcium pool activated by 20 Hz stimulation was explored by pharmacological manipulations of L-type calcium channels and the well-known ryanodine receptors in endoplasmic reticulum, which respond to cytoplasmic calcium by releasing more calcium [45,46,47].

As expected, blockade of L-type channels with nimodipine nearly abolished the fast calcium transient on 20 Hz stimulation (Figure 7). A second source of calcium produces the supralinear increase in the amplitude of the fast calcium transient. Blockade of calcium-induced calcium release by incubation of neurons with a cocktail of ryanodine (to block ryanodine-sensitive intracellular calcium channels) and thapsigargin (to deplete the internal calcium pool in the endoplasmic reticulum), reduced the fast calcium transient by ~40% [40]. Additional evidence for such phenomenon comes from the laboratories of Ludwig and Rice, showing that release of calcium from intracellular stores takes an active role in the release of peptides from magnocellular neurons [47] and the somatodendritic release of dopamine from nigrostriatal neurons [48]. Our experiments shown in Figure 7 then showed that the amplitude of the fast calcium transient determines the amount of somatic fluorescent spots, and clarified how the neuronal firing frequency determines the amount of exocytosis. As the firing frequency increases, temporal summation of calcium transients on each impulse sum and increase the amplitude of the fast calcium transient, thus activating calcium-induced calcium release. The final amplitude of the fast calcium transient determines the amount of exocytosis [40].

The intriguing observation that the fast calcium transient decays long before the arrival of vesicles at the plasma membrane along with that of vesicles remaining at resting positions when transmembrane calcium entry is abolished by extracellular magnesium, allowed us to put forward a new hypothesis: the primary role of the calcium tsunami is to set in motion the transport of vesicle clusters to the plasma membrane. By imaging fluorescence mitochondrial specific calcium sensor Rhod-2, we confirmed that the number of transported vesicle clusters depends sigmoidally on the stimulation frequency through the amount of calcium that invades the mitochondria (del Pozo-Sánchez, V., Mendez, B., and De-Miguel F.F., in preparation). Then we found that blocking calcium entry to the mitochondria prevents exocytosis. Therefore, newly synthesized ATP seems to provide the energy for kinesin-tubulin and actin myosin motor systems to carry vesicle clusters to the plasma membrane. Such transport was studied by using thermodynamic theory applied to the kinetics of the exocytosis, seen as the development of the fluorescent FM1-43 spots. The transport place at 15–90 nm/sec velocity, depending primarily on the vesicle cargo [26].

## 7. Exocytosis Is Sustained by a Positive Feedback Loop

A comparison of the kinetics of the calcium signals and exocytosis showed another striking result: the calcium signal that maintains the large-scale exocytosis is the peripheral transient. The kinetics of exocytosis showed that the lag to the onset of exocytosis reflects the distance and velocity of the vesicle cluster transport; the dynamic interval (a and a’ in Figure 8B) of the rate of exocytosis, which is orders of magnitude faster than the vesicle transport, reflects the velocity of the transport; the plateau (c and c’ in Figure 8) is reached at the end of exocytosis [26]. By assuming that each vesicle fusion contributes equally to the fluorescence increase, the amplitude of the transient reflects how many vesicles fused in response to a stimulation train [26].

It is common that exocytosis from a vesicle cluster is followed by exocytosis from a second vesicle cluster that arrives at the same membrane spot, thus producing a second florescence increase (a’ and c’ in Figure 8) [26,40]. Having said so, simultaneous optical recordings of the kinetics of exocytosis and calcium from individual release sites were used to analyze the relationships between exocytosis and the peripheral calcium transient.

Figure 8B shows that each FM fluorescence increase develops in parallel to an increase in the peripheral calcium transient. Moreover, each plateau in the exocytosis kinetics correlates with a calcium peak (b and b’ in Figure 8B). The cycles in the phase–state diagram in Figure 8C indicate a positive feedback loop established between calcium and exocytosis.

## 8. Serotonin That Has Been Released Activates the Third Calcium Pool

The immediate possibility that the third pool of calcium enters across the plasma membrane through voltage gated channels, ionotropic receptors, or through constitutive or capacitive channels which may evoke release in gland cells [49,50] was cancelled by the absence of any transmembrane current in voltage clamp experiments [40]. Instead, the calcium that promotes exocytosis originates from serotonin that has been released, first as result of constitutive exocytosis, later upon the large-scale exocytosis [40].

Figure 9A shows that application of serotonin with iontophoretic pipette to a spot of plasma membrane reproduces a peripheral calcium signal resembling that occurring during exocytosis. Such calcium elevation is not enough to evoke vesicle transport, presumably because it does not spread to internal cytoplasmic regions densely-populated with resting vesicles and the mitochondria that produces ATP for vesicle transport.

In simultaneous optical recordings of calcium and exocytosis, both signals are abolished by the serotonin antagonist methisergide, which has higher specificity for 5-HT2 receptors (Figure 9C). Activated 5-HT2 receptors are coupled to the release of intracellular calcium though activation of phospholipase C (PLC) and production of IP3. Both, exocytosis and the calcium elevation are abolished by the PLC blocker U-73122 (Figure 9D). Therefore, the positive feedback loop that maintains the large-scale exocytosis of serotonin incorporates a four-step sequence: (1) Serotonin that has been released activates 5-HT2 autoreceptors. (2) Activated 5-HT2 receptors activate PLC and IP3 production. (3) Active IP3 receptors release calcium from peripheral endoplasmic reticulum. (4) Calcium promotes exocytosis as vesicles dock with the plasma membrane. The feedback loop gets to an end when the last vesicles in a cluster fuse, serotonin is removed from the extracellular space, and calcium returns to resting levels. Such a mechanism has more general relevance, since in spite of lack of exploration in multiple neuron types, similar feedback loops maintain large-scale somatic exocytosis of serotonin by neuroepithelial cells [34], and release of peptides by magnocellular hypothalamic neurons [51] and from dorsal root ganglion neurons [52].

## 9. The Possible Calcium Sensors for Somatic Exocytosis

The mechanism for extrasynaptic exocytosis of serotonin by Retzius neurons is resumed in Figure 10. Calcium that is released from the endoplasmic reticulum is not in close proximity to the vesicles to fuse, as it is at synapses [53,54,55,56]. Such mismatch requires the inefficient intracellular calcium diffusion to reach docked vesicles. However, the levels of calcium that promote the release of peptides from the soma of dorsal root ganglion neurons is ten-fold lower than at synapses [1]. For that reason, extrasynaptic vesicles must carry high affinity calcium sensors different from the synaptotagmins 1 or 2 of synapses [57,58]. An interesting alternative has been found in the soma and dendrites of dopaminergic neurons by the laboratories of Rice and Trudeau [59,60]. Specific antibodies identify the presence of the highly sensitive isoforms 4 and 7 of the calcium sensor synaptotagmin.

## 10. On the High Energy Cost of the Use of Three Calcium Pools

The Swiss-clock precision of the machinery for somatic exocytosis imposes a high energy cost to the transport process. The amplified electron micrograph in Figure 11 shows details of the cytoarchitecture for somatic exocytosis. At rest or in neurons stimulated at 1 Hz, vesicles are associated to microtubules that connect to the plasma membrane [26,61]. Figure 11B shows that after 20 Hz stimulation, vesicles have made their way to the plasma membrane by displacing endoplasmic reticulum. In such a densely packed environment, the calcium-releasing endoplasmic reticulum, so necessary for the vesicle transport and large-scale exocytosis, hampers the vesicle passage to the plasma membrane. Friction forces then reduce the thermodynamic efficiency of the ATP usage and imposes a high energy cost to somatic exocytosis [61].

The ATP cost per vesicle and the thermodynamic efficiency of the vesicle transport have been calculated by applying the thermodynamic theory to the kinetics of exocytosis, which, as mentioned before, reports on the number of vesicles in the cluster, their travelling distance and velocity [26,61]. Internal vesicle clusters (red labels in Figure 11) are mobilized by kinesin coupled to microtubules; distant vesicle clusters already inside the actin cortex (green er labels in Figure 11) use myosin that is carried within the clusters to assembly a second transport system upon coupling to the actin cortex. It is remarkable that the vesicle clusters also carry the fuel-generator mitochondria all the way through to the cell periphery (Figure 11B).

The barrier imposed by the actin cortex to vesicle transport at rest turns into a permissive pathway in response to stimulation. Such an effect shown first in endocrine cells [62,63] appears repeatedly in neurons [21,64]. Evidence principally from endocrine cells points to calcium as a mediator of such transitions [65,66,67], although it has not yet been studied in relation to somatic exocytosis.

The ATP cost per vesicle and the thermodynamic efficiency of the ATP expenses are highly dependent on the traveling distance of the vesicles [26,61]. The slopes of the curves in Figure 12 correlate well with the topology of the traveling pathway. The thermodynamic efficiency of the ATP expenses correlates with the resting position of vesicle clusters, their size and traveling velocity (Figure 12B). Data allows for the prediction of four groups of vesicle clusters, each correlating with one slope of the ATP expenses and efficiency plots (Figure 12C) [61]. Group 1 (clear green) contains the most peripheral clusters, which display the highest thermodynamic efficiency values. As seen Figure 12C, such are the smaller clusters and rest immersed between both layers of endoplasmic reticulum. Their reduced size and their position near the plasma membrane suggest that they are the vesicle sources for constitutive release. The clusters in Group II (dark green) are embedded within the second layer of endoplasmic reticulum. Their transport velocities and efficiencies are smaller than those in Group I. The clusters in Group III (brown) rest in the borders of the actin cortex. Such large clusters move with high velocity and reach thermodynamic efficiencies above 1%, presumably because they are propelled by both types of motors. The clusters in Group IV (red) are large and have not yet entered the actin cortex. Their velocities are smaller than those in Group III presumably for being carried purely by tubulin kinesin transport [60].

It is to be noted that the 6.4% largest thermodynamic efficiency in Figure 12 is only one third of the ~20% efficiency of combustion motors of modern cars running on highways and one half the efficiency of the calcium ATPase [68]. It is remarkable that the modulation of the workings of the nervous system requires such high energy cost, paradoxically due to the sophisticated and highly regulated machinery that couples electrical activity to exocytosis.

## 11. Summary

Somatic exocytosis by Retzius neurons is regulated by a sequential activation of three calcium sources. The first two sources cooperate to regulate the amount of transmitter transported to the plasma membrane; the third source activates vesicle fusion.Increases in the frequency of electrical activity allow for the summation of transmembrane calcium transients generated upon activation of L-type calcium channels. As the stimulation frequency increases, summation increases the free calcium concentration and activates a second calcium source: calcium-induced calcium release.Calcium-induced calcium release increases the amplitude of the intracellular calcium transient. A resulting calcium tsunami invades the mitochondria and makes them produce ATP for the vesicle transport to the plasma membrane. The amplitude of the calcium transient in the mitochondria determines the number of vesicles transported and fused.Serotonin that is released maintains a large-scale exocytosis for hundreds of seconds through a third calcium source: the activation of 5-HT2 receptors coupled to phospholipase C and production of IP3 produces calcium release from endoplasmic reticulum adjacent to the plasma membrane.The calcium-releasing endoplasmic reticulum and the actin cortex generate obstacles to the transport pathway of vesicles. The resulting friction forces reduce the thermodynamic efficiency of the ATP usage, adding energy cost to somatic exocytosis.

## Figures and Tables

**Figure 1 ijms-23-01495-f001:**
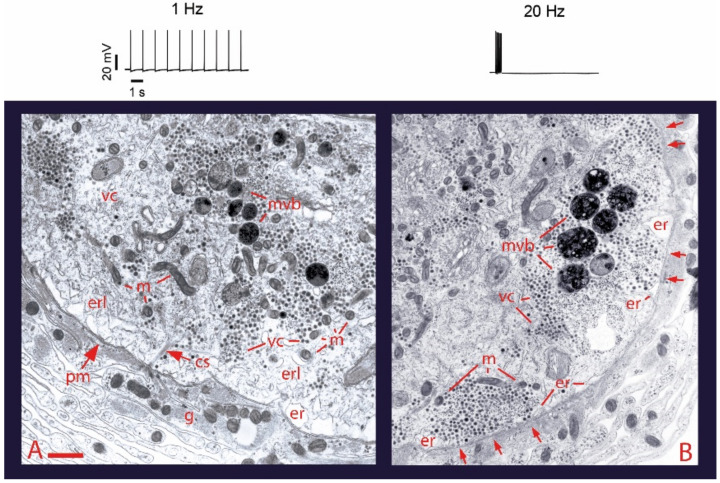
Electrical activity promotes vesicle transport to the plasma membrane. (**A**) Electron micrograph of a soma fixed after 1 Hz stimulation (train on top). Vesicle clusters remain distant from the plasma membrane. (**B**) After 20 Hz stimulation (top) vesicle clusters appear apposed to the plasma membrane (arrows). vc = vesicle clusters; mt = microtubules; pm = plasma membrane; m = mitochondria; er = endoplasmic reticulum; erl = internal endoplasmic reticulum layers; mvb = multivesicular bodies; g = glial cell processes. Scale bar = 1 µm. Obtained with permission from [26].

**Figure 2 ijms-23-01495-f002:**
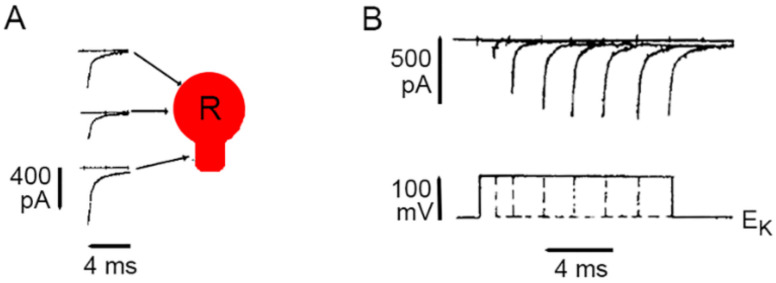
Calcium currents though L-type channels in the soma membrane. (**A**) Loose patch clamp recordings of calcium tail currents from small patches of membrane upon 10 ms depolarization. All recordings were obtained with the same pipette, thus making the amplitude of currents comparable for reflecting the density of active channels. (**B**) Lack of inactivation of somatic calcium current indicating predominant activation of L-type channels. The activation kinetics were reconstructed by superimposing tail currents upon depolarizations with increasing durations. Potassium tail currents were prevented by clamping the voltage at the potassium equilibrium potential (E_K_). Sodium tail currents were eliminated by reducing the sodium concentration in the pipette. Adapted with permission from the publisher of [39].

**Figure 3 ijms-23-01495-f003:**
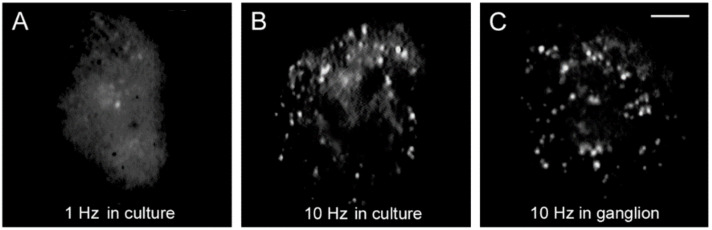
Somatic exocytosis as measured from FM1-43 fluorescence staining. (**A**) The fluorescent FM1-43 spots after stimulation with a 1 Hz train correspond to constitutive exocytosis. (**B**) Increase in the number of fluorescent spots after a 10 Hz train. (**C**) Similar staining pattern in the soma of a neuron stimulated in the ganglion. Scale bar = 10 μm. Obtained with permission from [32].

**Figure 4 ijms-23-01495-f004:**
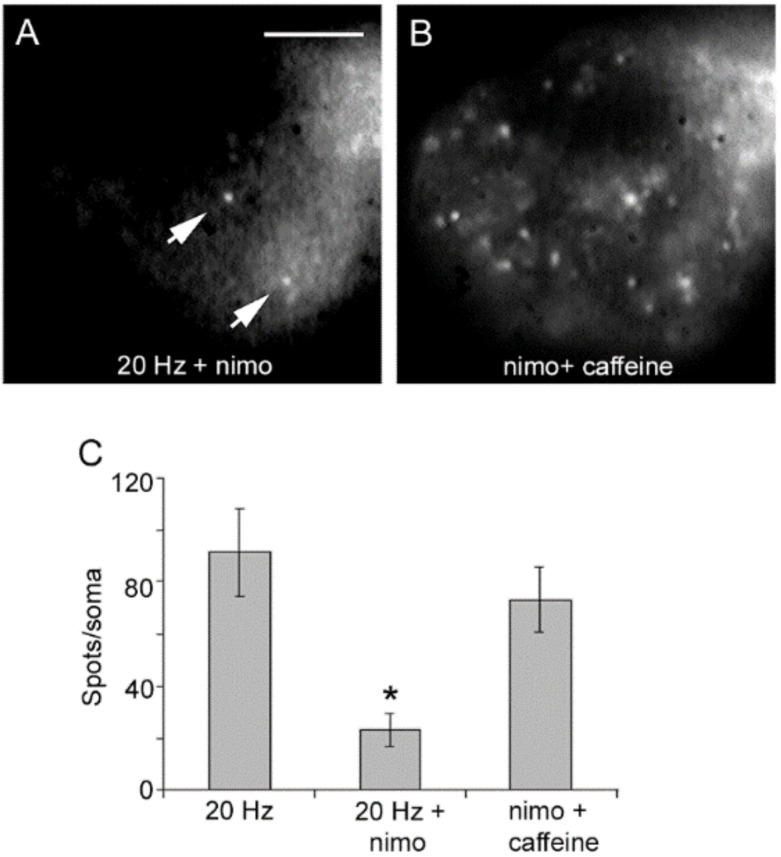
Activation of L-type calcium channels mediate somatic exocytosis. (**A**) Incubation with nimodipine (nimo) abolishes evoked but not constitutive exocytosis upon 20 Hz stimulation. The spots pointed by the arrow remained in B. (**B**) Caffeine in the presence of nimodipine evokes vesicle transport and exocytosis in the absence of electrical stimulation. Images are selected areas of the soma. (**C**) Exocytosis quantified as the number of fluorescent FM1-43 spots per soma. Bars are the S.E.M.; the * indicates a *p* < 0.05 T test significance with respect to the 20 Hz and nimo+caffeine groups. The 20 Hz and nimo+caffeine groups were similar (*p* > 0.05). Reproduced with permission from the publisher of [32].

**Figure 5 ijms-23-01495-f005:**
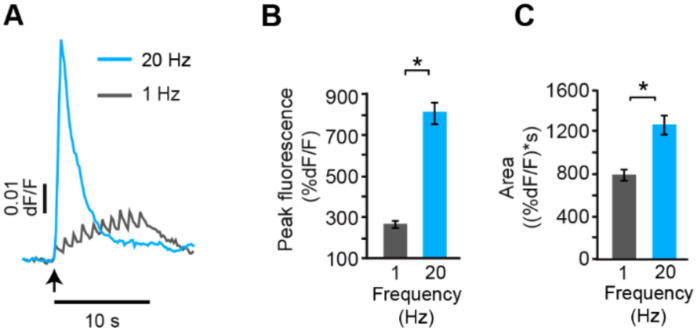
Frequency dependence of calcium signals. (**A**) Calcium transients at the vicinity of the plasma membrane. (**B**) Peak amplitude of the transients. (**C**) Integral of the calcium transients. The supralinear summation of calcium upon 20 Hz stimulation seen from the difference between the amplitude and the integral of the transients suggest an additional calcium component. The * indicates *p* < 0.05 significant differences. Obtained with permission from [40].

**Figure 6 ijms-23-01495-f006:**
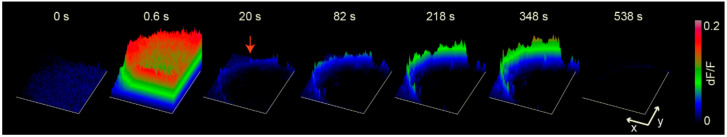
Calcium dynamics on 20 Hz stimulation. Time-lapse sequence of fluorescent images showing the fast calcium transient invading the soma followed by the persistent calcium transient in the periphery (red arrow). Stimulation occurred at time = 0 s. The time after stimulation is indicated above each snapshot. The color code indicates the amplitude of the normalized (dF/F) fluorescence of the calcium sensor. Obtained with permission from [40].

**Figure 7 ijms-23-01495-f007:**
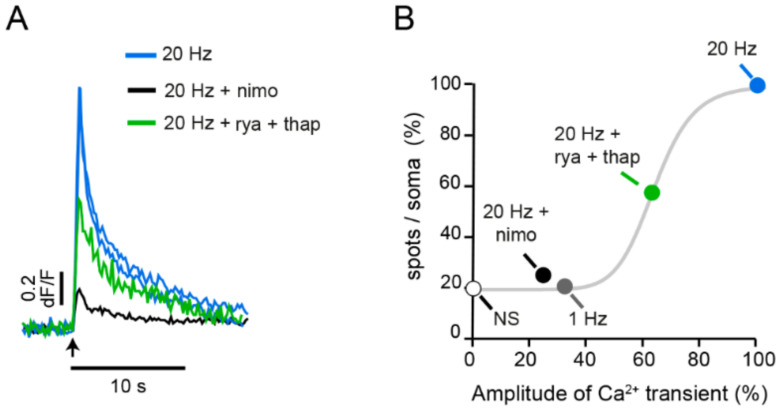
The fast calcium transient determines the amount of exocytosis. (**A**) Kinetics of the fast calcium transient imaged besides the plasma membrane. Application of L-type calcium blocker nimodipine reduces the transient by 80%. Elimination of calcium-induced calcium release by combining ryanodine and thapsigargin reduces the calcium transient by 40%. (**B**) The amplitude of the fast calcium transient determines the amount of release. The grey sigmoidal is the frequency-dependence on the amount of exocytosis. Constitutive exocytosis contributes with ~20% of the spots per soma. NS = on stimulated. Obtained with permission from [40].

**Figure 8 ijms-23-01495-f008:**
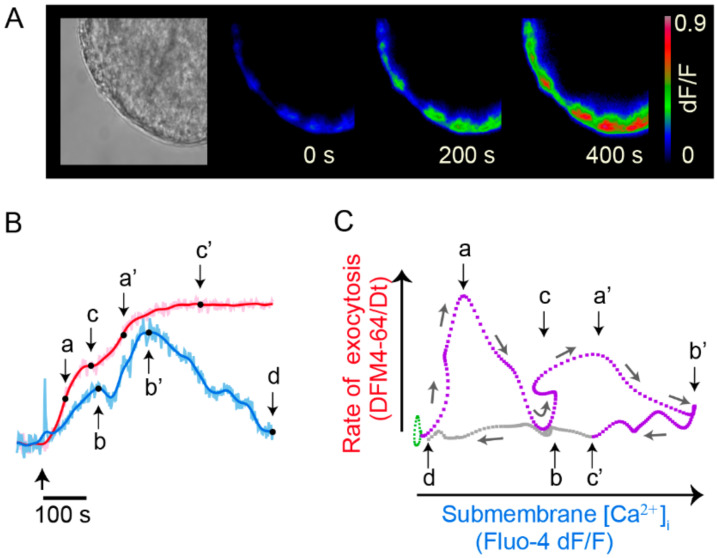
Exocytosis is maintained by a positive feedback loop. (**A**) Slow development of FM4-64 dye fluorescent spots during exocytosis upon 20 Hz train. Images were taken from spots in an equatorial plane of the soma in culture. (**B**) Kinetics of exocytosis (red) and calcium (blue) from simultaneous optical recordings. Exocytosis displays two sigmoidal steps, each corresponding to a vesicle cluster. a and a’ are the dynamic ranges of exocytosis; c and c’ are the plateaus indicating the end of exocytosis from each vesicle cluster. In the calcium signal the fast transient appears as a spike synchronized to stimulation (thick arrow). Following is the peripheral transient with two elevations, each correlating with an exocytosis bulk. The peaks of the calcium signal and b’ occur by the end of exocytosis. When exocytosis finishes the calcium levels return to rest. (**C**) State–phase diagram for the kinetics in C, indicating feedback loops between calcium and exocytosis. Each cycle reflects exocytosis from one vesicle cluster. Reproduced with permission from [40].

**Figure 9 ijms-23-01495-f009:**
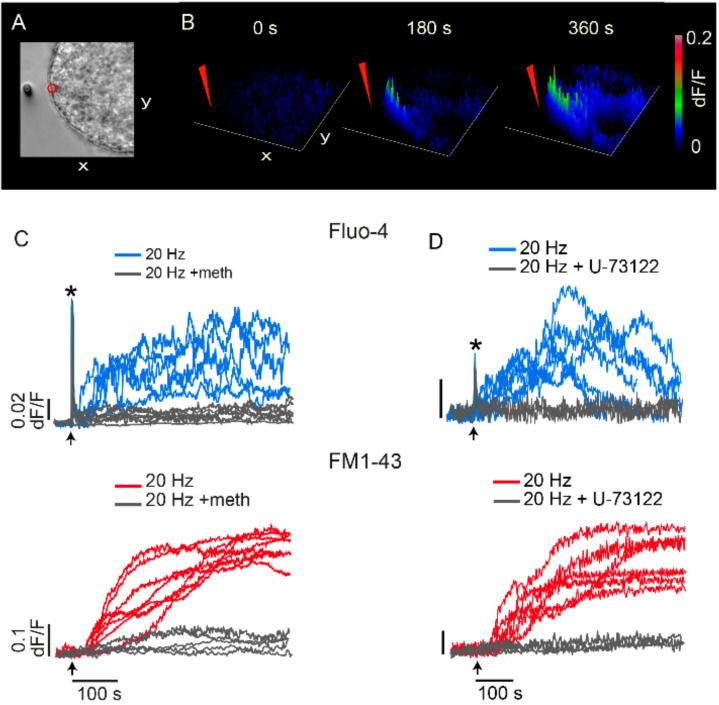
Activation of serotonin receptors and PLC are needed for the feedback loop. (**A**) Soma of cultured neuron and the tip of pipette for iontophoretic serotonin application (left). The region from which calcium measurements were made is circled in red. (**B**) Calcium elevations upon serotonin application in the neuron in A. (**C**) Methisergide abolishes the peripheral calcium elevation (top left) and exocytosis (bottom left). (**D**) The PLC blocker U-73122 abolishes calcium elevations (top right) and exocytosis (bottom right). The * in C and D indicate the fast calcium transient, which appear truncated due to the image acquisition rate. The x and y axis in A and B correspond to 30 μm. Adapted from [40].

**Figure 10 ijms-23-01495-f010:**
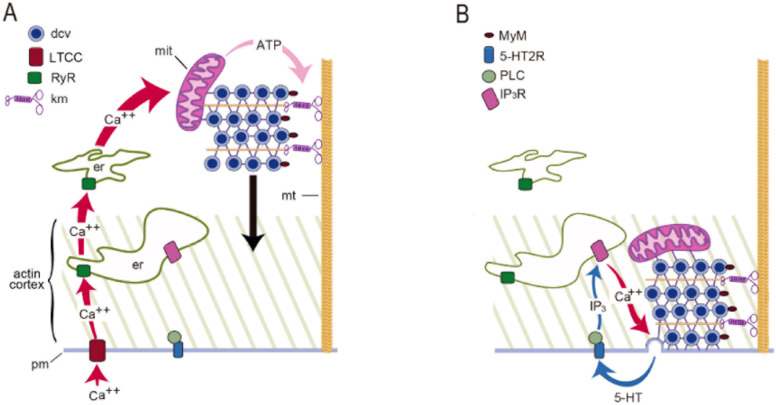
Mechanism for somatic exocytosis of serotonin. (**A**) A train of action potentials produces opening of L-type calcium channels (LTCC). Calcium entry activates ryanodine receptors (RyR) in endoplasmic reticulum (er) to produce calcium-induced calcium release. The amplified calcium wave invades the soma. In the mitochondria (mit) calcium stimulates ATP synthesis. ATP activates kinesin motors (km) that transport dense core vesicles (dcv) along microtubules (mt). The black arrow indicates the vesicle transport. (**B**) As vesicles enter the actin cortex, myosin motors (MyM) couple to actin and contribute to transport. Release is maintained by a positive feedback loop. Serotonin that is released activates 5-HT2 receptors (5-HT2R). Activation of phospholipase C (PLC) produces IP3, which binds to receptors (IP3R) to activate intracellular calcium release. Such calcium maintains exocytosis going on until last vesicles fuse. Obtained with permission from [22].

**Figure 11 ijms-23-01495-f011:**
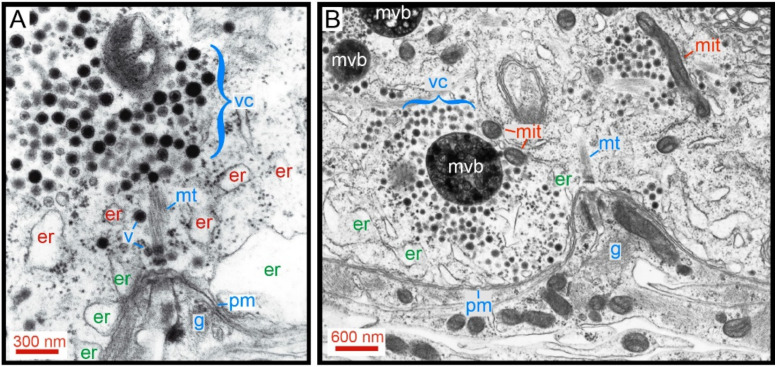
Cytoarchitecture of the vesicle transport pathway. (**A**) Amplified electron micrograph of a soma fixed after 1 Hz stimulation. A somatic vesicle cluster (vc) is bound to the plasma membrane (pm) though microtubules (mt). Peripheral endoplasmic reticulum (red er) rest apposed to the plasma membrane. A more internal layer of endoplasmic reticulum (green er) contains smaller bags. Individual vesicles (v) from peripheral clusters approach the plasma membrane along microtubules suggesting their contribution to constitutive exocytosis. Glial cell processes (g) penetrate an invagination of the Retzius neuron where microtubules anchor. (**B**) Electron micrograph of a release site in a neuron that had been stimulated with 20 Hz trains. Vesicles from a large cluster arrived at the plasma membrane after passing through a bottleneck of endoplasmic reticulum. Mitochondria (mit) travel around vesicle clusters. Multi-vesicular bodies (mvb) form upon endocytosis. Note that exocytosis occurs onto glia. Obtained with permission from [61].

**Figure 12 ijms-23-01495-f012:**
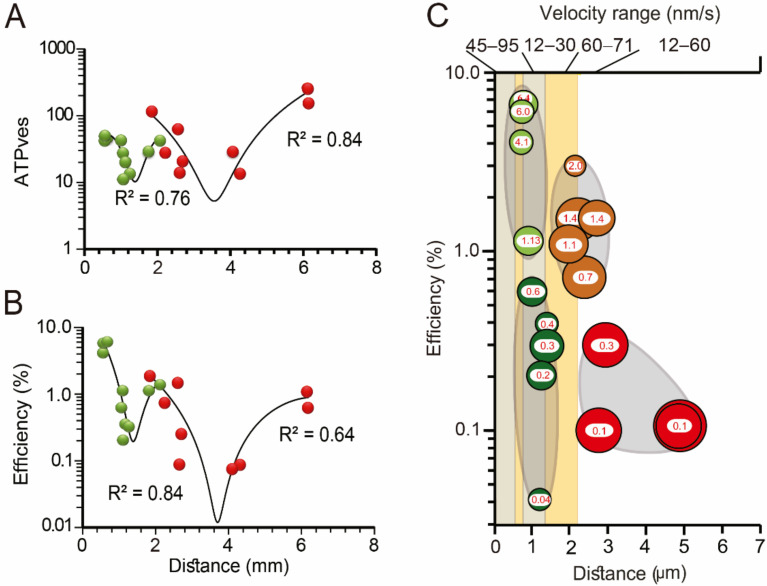
Structural and biophysical correlates of the thermodynamic efficiency of the ATP hydrolysis during vesicle transport. The ATP cost per vesicle fused (**A**) and the thermodynamic efficiency (**B**) depend on the travelling distance of the vesicle clusters. Distances were calculated from model fittings to the exocytosis kinetics in combination with electron microscopy measurements. (**C**) Peripheral (green) and internal (red) vesicle clusters are grouped according to their thermodynamic efficiency, size, distance from the plasma membrane and transport velocity. The thermodynamic efficiency of somatic exocytosis (% inside vesicle clusters) correlated with the thickness of the actin cortex (pale orange) and with the position of both layers of endoplasmic reticulum (gray bars). The cluster diameters are proportional to the number of vesicles that fused in each cluster. Adapted with permission from [61].

## Data Availability

No new data were created or analyzed in this study. Data sharing is not applicable to this article.

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
