# Peer review of "The Thermodynamically Expensive Contribution of Three Calcium Sources to Somatic Release of Serotonin"

_ijms, 2022, doi:10.3390/ijms23031495_

Round 1

Reviewer 1 Report

The review by Francisco F. De-Miguel entitled “The Thermodynamically Expensive Contribution Of Three Calcium Sources To Somatic Release Of Serotonin” aims to summarize the existing knowledge on cooperative actions of three calcium sources on somatic exocytosis of serotonin. The topic of the present review is interesting because it concerns unconventional extra synaptic release of neurotransmitters.

There are several weaknesses in this review.

First, it focuses on the somatic release of neurotransmitters from serotonergic Retzius neurons, whereas the data from other types of neuronal cells are largely not included.

Second, the three sources of calcium are poorly defined in the paper. While the stimulatory role of the L-type calcium channels is adequately described, it is not clear what are the other calcium sources in the context of this review. Author describes data obtained with caffeine, ryanodine, thapsigargin, and serotonin. However, from the review it is not clear whether functionally separate intracellular calcium stores serve as the separate sources of calcium.

Third, author does not mentioned the role of non-voltage-gated calcium influx, which recently emerged as powerful modulator of vesicle recruitment to the releasable pool (see for example Cheek and Thorn, Cell Calcium, 2006, 40: pp. 309-318;  Yang and Fomina,  Cell Calcium 2020, 87:102184). Finally, the reviewed papers are somewhat old; out of 58 articles cited, only 11 articles are published after 2015 and 19 after 2010. Other specific flaws in the paper are listed below. Overall, after revision, this review will be of interest to the researchers in the fields of neuroscience and cell biology.

Specific comments:

  1. Wherever the author presents original data, it would be appropriate to mention the names of the authors of these data (e.g. “De-Miguel and colleagues have shown that” or “ a study from the Neher’s lab demonstrated that”)
  2. Lines 51-53: It is not clear what author wants to say in this paragraph. First sentence mentions 3 complementary roles of calcium ions, whereas second and third paragraphs refer to 3 calcium sources. Please, clarify.
  3. Line 54: The first figure mentioned in the manuscript is Figure 3, instead of Figure 1.
  4. Line 80: What are the “La type channels” ?
  5. Line 95: correct Figure 3B number.
  6. Line 109: Correct spelling of “sport”
  7. Line 125- 128: Mechanism of the caffeine-induced exocytosis needs to be discussed.
  8. Figures: Author should indicate whether permissions for reproduction of figures were obtained from the publishers of original papers.
  9. Figure 1: Not all abbreviations present in the images are explained in the figure legend (e.g. erl, cs). Reference to the original paper should be provided in the body of the manuscript.
  10. Figure 3. reference to Figure 3 is missing in the text.
  11. References: Autor uses different reference formats
  12. Reference # 27 Del-Pozo et al. – Paper in preparation for submission should be removed from the list of references.

Author Response

Dear Reviewer, I wish to acknowledge the careful reading to my paper. The suggestions and corrections have been considered in the present version. Following are the explanations about how each issue has been taken care of.

Reviewer: First, it focuses on the somatic release of neurotransmitters from serotonergic Retzius neurons, whereas the data from other types of neuronal cells are largely not included.

Author: The references had been included in each section. However, they are more explicit now by stating the particular cell types in which each phenomenon has been detected.

It is to be noted that several of our observations are original and have not been repeated elsewhere. Such is the case of the mitochondrial synthesis of ATP, the measurement of the slow peripheral calcium transient and the thermodynamic cost and efficiency.  

Following are sentences with references to different neuron types having the same mechanisms described here:

  1. The paragraph starting in Line 90 contains multiple examples of L and N calcium channels and glutamate receptors mediating calcium entry in different types of neurons.
  2. Line 136 reads: “A plateau in the amount of release at 20 Hz in Retzius neurons, in lobster serotonergic neurons and in mammalian peptidergic neurons,41-43 suggests a generalyzed peak of efficiency for the coupling between electrical activity, calcium and exocytosis.”
  3. Line 225 now reads: “Additional evidence for such phenomenon comes from the laboratories of Ludwig and Rice, showing that release of calcium from intracellular stores takes an active role on release of peptides from magnocellular neurons48 and somatodendritic release of dopamine from nigrostriatal neurons.49”
  4. Line 322 reads: “Such mechanism has more general relevance, since in spite of lack of exploration in multiple neuron types, similar feedback loops maintain large-scale somatic exocytosis of serotonin by neuroepithelial cells,35 and release of peptides by magnocellular hypo-thalamic neurons52 and from dorsal root ganglion neurons.53
  5. Line 334 reads: “An interesting alternative has been found in the soma and dendrites of dopaminergic neurons by the laboratories of Rice and Trudeau.60,61 Specific antibodies identify the presence of the highly sensitive isoforms 4 and 7 of the calcium sensor synaptotagmin.”

Reviewer: Second, the three sources of calcium are poorly defined in the paper. While the stimulatory role of the L-type calcium channels is adequately described, it is not clear what are the other calcium sources in the context of this review. Author describes data obtained with caffeine, ryanodine, thapsigargin, and serotonin. However, from the review it is not clear whether functionally separate intracellular calcium stores serve as the separate sources of calcium.

Author: This issue has been corrected from the abstract, which from Line 13 now reads:

“Trains of action potentials induce transmembrane calcium entry through L-type channels. For action potential frequencies above 5 Hz, summation of calcium transients on individual action potentials activates the second calcium source: ryanodine receptors produce calcium–induced calcium release. The resulting calcium tsunami activates mitochondrial ATP synthesis to fuel transport of vesicles to the plasma membrane. Serotonin that is released maintains a large-scale exocytosis by activating the third calcium source: serotonin autoreceptors coupled to phospholipase C promote IP3 production. Activated IP3 receptors in peripheral endoplasmic reticulum release calcium that promotes vesicle

Reviewer: Third, author does not mentioned the role of non-voltage-gated calcium influx, which recently emerged as powerful modulator of vesicle recruitment to the releasable pool (see for example Cheek and Thorn, Cell Calcium, 2006, 40: pp. 309-318;  Yang and Fomina,  Cell Calcium 2020, 87:102184).

 Author: This along with the references suggested by the reviewer appear in Line , where we also mention that voltage clamp experiments did not show any transmembrane calcium entry. Later in the paper it becomes clear that exocytosis and calcium can be eliminated by dealing directly with serotonin receptors and second messengers.

Line 294 reads: “The immediate possibility that the third pool of calcium enters across the plasma membrane through voltage gated channels, ionotropic receptors, or through constitutive or capacitive channels which may evoke release in gland cells50, 51 was cancelled by the absence of any transmembrane current in voltage clamp experiments.41  

Reviewer: Finally, the reviewed papers are somewhat old; out of 58 articles cited, only 11 articles are published after 2015 and 19 after 2010. Other specific flaws in the paper are listed below. Overall, after revision, this review will be of interest to the researchers in the fields of neuroscience and cell biology.

 Author: Several new papers published after 2015 have been added to this version of the paper.

 Reviewer: Specific comments:

Wherever the author presents original data, it would be appropriate to mention the names of the authors of these data (e.g. “De-Miguel and colleagues have shown that” or “ a study from the Neher’s lab demonstrated that”)

Author: We are following such suggestion in several paragraphs along the paper. For example, Line 70 states:

“Our laboratory has also found that after 20 Hz stimulation in the presence of extracellular magnesium to block transmembrane calcium entry vesicles remain at resting positions (Del Pozo-Sánchez, V., Mendez, B. and De-Miguel, F.F., in preparation)..”

Line 225 states: “Additional evidence for such phenomenon comes from the laboratories of Ludwig and Rice, showing that release of calcium from intracellular stores takes an active role on release of peptides from magnocellular neurons48 and somatodendritic release of dopamine from nigrostriatal neurons.49 Our experiments shown in Figure 7 then showed that the amplitude of the fast calcium transient determines the amount of somatic fluorescent spots,”

Line 334 states: “An interesting alternative has been found in the soma and dendrites of dopaminergic neurons by the laboratories of Rice and Trudeau.60,61

Reviewer: Lines 51-53: It is not clear what author wants to say in this paragraph. First sentence mentions 3 complementary roles of calcium ions, whereas second and third paragraphs refer to 3 calcium sources. Please, clarify.

Author: The paragraph was confusing. Therefore it has now been changed from the title, according to a suggestion of Reviewer 2. The title in Line 57 now reads:

“Electrical activity and calcium promote vesicle transport and somatic exocytosis”

Line 59: “Ultrastructural analysis of the soma of Retzius neurons under different stimulation frequencies permits an initial understanding on the role of electrical stimulation and calcium on somatic exocytosis. “

The sections mentioning the three sources of calcium start in Line 88, and each following section defines the sources of calcium.

Reviewer: Line 54: The first figure mentioned in the manuscript is Figure 3, instead of Figure 1.

Author: Corrected.

Reviewer: Line 80: What are the “La type channels” ?

Author: Corrected to L-type channels

Reviewer: Line 95: correct Figure 3B number.

Author: Corrected to Figure 2B

Reviewer: Line 109: Correct spelling of “sport”

Author: Corrected to spot

Reviewer: Line 125- 128: Mechanism of the caffeine-induced exocytosis needs to be discussed.

Author: Line 164 now explains:

“Subsequent addition of caffeine to such same neurons to promote release of intracellular calcium through ryanodine receptors44-47 in absence of electrical stimulation, produced a bulk of exocytosis, seen as an increased number of fluorescent spots (Figure 4B). Such results clarified that calcium entry upon activation of L type channels promotes both, vesicle transport and exocytosis. Moreover, they suggested that calcium-induced calcium release is part of the link between excitation and exocytosis.”

Reviewer: Figures: Author should indicate whether permissions for reproduction of figures were obtained from the publishers of original papers.

Author: Done for all figures, although some of the Figure rights belong to the author.

Reviewer: Figure 1: Not all abbreviations present in the images are explained in the figure legend (e.g. erl, cs). Reference to the original paper should be provided in the body of the manuscript.

Author: Done

Reviewer: Figure 3. reference to Figure 3 is missing in the text.

Author: It has now been added in Line 132.

Reviewer: References: Autor uses different reference formats

Author: Corrected.

Reviewer: Reference # 27 Del-Pozo et al. – Paper in preparation for submission should be removed from the list of references.

Author: Done

Reviewer 2 Report

Overview and general recommendation:

In this manuscript, the author reviewed the somatic release of serotonin through the extrasynaptic mechanism (in the absence of synaptic structures), with a focus on the three calcium sources involved, namely transmembrane calcium entry through L-type channels activated by action potentials, calcium–induced calcium release (CICR) activated by elevated the intracellular calcium level and IP3 induced calcium release from IP3 receptors on the ER membrane. The review is supported by solid experimental data previously published by the author's lab, which illustrates the significant contribution of the author's lab to the field.

Major comments:

  1. In the abstract, I would suggest making it more evident to the readers the difference between the three calcium resources, i.e. the first two calcium sources cooperate to induce the transport of vesicles, while the third one catalyzes vesicle fusion.
  2. Reproduction of each previously published figure should obtain permission from the publisher separately.
  3. The structure of the review could be further optimized. For example, the first section seems to be a general introduction to the full review. It would be nice to do minor modifications to help it serve this role, and the title might be deleted. Under the section title of "Three Calcium Sources Cooperate To Somatic Exocytosis", the content does not correspond well with the title. It may also be helpful to construct the main body of the review based on the three calcium resources and the energy cost of the process and add subtitles when needed.
  4. The readers would appreciate a final summary/conclusion paragraph to provide critical take-home information.

Minor comments:

  1. Line 48: The "crawling circuit" needs to be briefly explained.
  2. Line 54: It should be Figure 1, not Figure 3
  3. Line 79: To say "lack of activation" is not entirely accurate because in the absence of a calcium-dependent mechanism, a voltage-dependent mechanism alone will still cause the slow inactivation of LTCC, and CaM can promote and accelerate calcium-dependent inactivation
  4. Lane 80: L-type calcium channels instead of La type channels? It may help to either use L-type calcium channel or LTCC, which is a commonly used abbreviation, throughout the manuscript (for example, LCach in line 299 is not commonly used)
  5. Line 95: It should be Figure 2B, not Figure 3B. Also, it would be clearer to define that the calcium current lacks inactivation under test conditions.
  6. Figure 3: It is recommended to label the cell types in Figures 3A and 3B in a similar way to Figure 3C.
  7. Line 146: It should be "C" instead "E" for the bar graph.
  8. Line 148: the asterisk should indicate P<0.05, not P>0.05. Also, statistical analysis results should be labelled between 20 Hz and 20 HZ+nino, between 20 HZ+nino and nino+caffeine and between 20 Hz and nino+caffeine, respectively.
  9. Line 150: This section used two elegant studies to show exemplar results from the analysis of calcium dynamics. However, the results should be preceded with an introduction of the known knowledge of calcium dynamics and followed by a conclusion from this section.
  10. Line 183: same as above. A general introduction to this topic referencing other people's work would help.
  11. Line 255: "5-HT2" should be consistently used throughout the review instead of 5HT2.
  12. Line 257: This is a bit confusing to call InsP3Rs "IP3 receptor-channels". It may be more accurate to introduce the IP3 receptor as a ubiquitously expressed Ca2+-release channel on the ER.
  13. The review needs to be carefully examined for typos.

Author Response

Dear Reviewer, I wish to acknowledge the careful reading to my paper. The suggestions and corrections have been considered in the present version. Following are the explanations about how each issue has been taken care of.

Reviewer: Major comments:

In the abstract, I would suggest making it more evident to the readers the difference between the three calcium resources, i.e. the first two calcium sources cooperate to induce the transport of vesicles, while the third one catalyzes vesicle fusion.

Author: Done. The abstract now reads (L13-20): “Trains of action potentials induce transmembrane calcium entry through L-type channels. For action potential frequencies above 5 Hz, summation of calcium transients on individual action potentials activates the second calcium source: ryanodine receptors produce calcium–induced calcium release. The resulting calcium tsunami activates mitochondrial ATP synthesis to fuel transport of vesicles to the plasma membrane. Serotonin that is released maintains a large-scale exocytosis by activating the third calcium source: serotonin autoreceptors coupled to phospholipase C promote IP3 production. Activated IP3 receptors in peripheral endoplasmic reticulum release calcium that promotes vesicle fusion.”

Reviewer: Reproduction of each previously published figure should obtain permission from the publisher separately.

Author: Done. It is to be noted that several publications from which Figures were adapted are of free access and the author of this review retains the reproduction rights. That was the reason for not making specific statements before. 

Reviewer: The structure of the review could be further optimized. For example, the first section seems to be a general introduction to the full review. It would be nice to do minor modifications to help it serve this role, and the title might be deleted.

Author: Done

Reviewer: Under the section title of "Three Calcium Sources Cooperate To Somatic Exocytosis", the content does not correspond well with the title. It may also be helpful to construct the main body of the review based on the three calcium resources and the energy cost of the process and add subtitles when needed.

Author: Done. The new title (L 64) now reads: “Electrical activity and calcium promote vesicle transport and somatic exocytosis.”

The first paragraph has been added with three introductory lines (L. 58-60), which read:

“Ultrastructural analysis of the soma of Retzius neurons under different stimulation frequencies permits an initial understanding on the role of electrical stimulation and calcium on somatic exocytosis..”  

Reviewer: The readers would appreciate a final summary/conclusion paragraph to provide critical take-home information.

Author: Done. A summary has been added in P. 11 L. 445:

Summary

  1. Somatic exocytosis by Retzius neurons is well-regulated by a sequential activation of three calcium sources. The first two cooperate to regulate the amount of transmitter transported to the plasma membrane; the third source activates vesicle fusion.
  2. Increases in the frequency of electrical activity allow summation of transmembrane calcium transients generated upon activation of L-type calcium channels. As the stimulation frequency increases, summation increases the free calcium concentration and activates a second calciun source: calcium-induced calcium release.
  3. Calcium-induced calcium release increases the amplitude of the intracellular calcium transient. A resulting calcium tsunami invades the mitochondria and make them produce ATP for the vesicle transport to the plasma membrane. The amplitude of the calcium transient in the mitochondria determines the number of vesicles transported and fused.
  4. Serotonin that is released maintains a large-scale exocytosis for hundreds of seconds through a third calcium source: activation of 5-HT2 receptors coupled to phospholipase C and production of IP3 produces calcium release from endoplasmic reticulum adjacent to the plasma membrane.

The calcium-releasing endoplasmic reticulum and the actin cortex generate obstacles to the transport pathway of vesicles. The resulting friction forces reduce the thermodynamic efficiency of the ATP usage, adding energy cost to somatic exocytosis.”

Reviewer: Minor comments:

Reviewer: Line 48: The "crawling circuit" needs to be briefly explained.

Author: Lines 52-62 now read: “For the moment it is useful to mention that electrical stimulation of Retzius neurons trains of 10 impulses at 20 Hz in an isolated ganglion with, activates the neuronal circuit that controls crawling locomotor behavior.24 During crawling, bursts of motoneuron firing produce alternate elongation of front body segments with shortening of rear segments.25,26 The minute range latency to produce crawling is due first, to the series of calcium-dependent processes (the matter of this review), that take place between electrical activity and serotonin liberation. Unpublished experiments by Sanchez-Sugía and the author of this review show that the second part of the long latency to behavior reflects the arrival time of released serotonin to distant neuron targets that codify for crawling (Sanchez-Sugía C., and De-Miguel, F.F., in preparation)..”

Reviewer: Line 54: It should be Figure 1, not Figure 3

Author: Corrected

Reviewer: Line 79: To say "lack of activation" is not entirely accurate because in the absence of a calcium-dependent mechanism, a voltage-dependent mechanism alone will still cause the slow inactivation of LTCC, and CaM can promote and accelerate calcium-dependent inactivation

Author: Corrected now in L 92

Reviewer: Lane 80: L-type calcium channels instead of La type channels?

Author: Corrected

Reviewer: It may help to either use L-type calcium channel or LTCC, which is a commonly used abbreviation, throughout the manuscript (for example, LCach in line 299 is not commonly used)

Author: Corrected in Figure 10 and in L339 of the present version.

Reviewer: Line 95: It should be Figure 2B, not Figure 3B. Also, it would be clearer to define that the calcium current lacks inactivation under test conditions.

Author: Corrected. Now Line 103.

Reviewer: Figure 3: It is recommended to label the cell types in Figures 3A and 3B in a similar way to Figure 3C.

Author: Done.

Reviewer: Line 146: It should be "C" instead "E" for the bar graph.

Author: Corrected now in L.168.

Reviewer: Line 148: the asterisk should indicate P<0.05, not P>0.05.

Author: Done. Now line 169.

Reviewer: Also, statistical analysis results should be labelled between 20 Hz and 20 HZ+nino, between 20 HZ+nino and nino+caffeine and between 20 Hz and nino+caffeine, respectively.

Author: Statistics have been added to the Figure legend, which now reads: the asterisk indicates a P<0.05 T test significance with respect to the 20Hz and nimo+caffeine groups. The 20Hz and nimo+caffeine groups were similar (P>0.05).

Reviewer: Line 150: This section used two elegant studies to show exemplar results from the analysis of calcium dynamics. However, the results should be preceded with an introduction of the known knowledge of calcium dynamics and followed by a conclusion from this section.

Author: A short introduction was added in Line 173, which now reads: Essential to understand how the firing frequency is translated into the amount of somatic release was to analyze the calcium elevations produced by different the stimulation frequencies.”

As a conclusion to the section, Line 201 now reads:

“Summarizing these observations, 20 Hz stimulation produces summation of the transmembrane calcium elevation that follows each action potential. Activation of a second calcium pool increases rapidly the concentration of the calcium wave and calcium invades the soma. A third calcium pool in the periphery correlates with the duration of exocytosis.”

Reviewer: Line 183: same as above. A general introduction to this topic referencing other people's work would help.

Author: Done. An introduction with references has been added in Line 213, which now reads:

“The possibility of calcium-induced calcium release being the second calcium pool activated by 20 Hz stimulation was explored by pharmacological manipulations of L-type calcium channels and the well-known ryanodine receptors in endoplasmic re-ticulum, which respond to cytoplasmic calcium by releasing more calcium.45-47

A conclusion paragraph had already been added to the end of the section, on Line 227:

“Our experiments shown in Figure 7 then showed that the amplitude of the fast calcium transient determines the amount of somatic fluorescent spots, and clarified how the neuronal firing frequency determines the amount of exocytosis. As the firing frequency increases, temporal summation of calcium transients on each impulse sum and increase the amplitude of the fast calcium transient, thus activating calcium-induced calcium release. The final amplitude of the fast calcium transient determines the amount of exocytosis.41 “  

Reviewer: Line 255: "5-HT2" should be consistently used throughout the review instead of 5HT2.

Author: Corrected in lines 310-318

Reviewer: Line 257: This is a bit confusing to call InsP3Rs "IP3 receptor-channels". It may be more accurate to this has been corrected in Line introduce the IP3 receptor as a ubiquitously expressed Ca2+-release channel on the ER.

Author: This has been corrected in Lines 310-318, which now read: “Activated 5-HT2 receptors activate PLC and IP3 production. 3. Active IP3 receptors release calcium from peripheral endoplasmic reticulum.”

Reviewer: The review needs to be carefully examined for typos.

Author: Done, thank you very much!

Round 2

Reviewer 1 Report

Author addressed all concerns indicated in the first review, which significantly improved the manuscript.